# Bioactive Peptides Obtained from Legume Seeds as New Compounds in Metabolic Syndrome Prevention and Diet Therapy

**DOI:** 10.3390/foods11203300

**Published:** 2022-10-21

**Authors:** Kaja Kiersnowska, Anna Jakubczyk

**Affiliations:** Department of Biochemistry and Food Chemistry, University of Life Sciences in Lublin, Skromna 8, 20-704 Lublin, Poland

**Keywords:** legume, bioactive peptides, metabolic syndrome, enzyme inhibitory

## Abstract

Currently, food is regarded not only as a source of nutrients, vitamins, and minerals but also as a source of bioactive compounds that can play a significant role in the prevention and diet therapy of many diseases. Metabolic syndrome (MS) is a complex disorder defined as a set of interrelated factors that increase the risk of cardiovascular disease, atherosclerosis, type 2 diabetes, or dyslipidemia. MS affects not only adults but also children. Peptides are one of the compounds that exhibit a variety of bioactive properties. They are derived from food proteins, which are usually obtained through enzymatic hydrolysis or digestion in the digestive system. Legume seeds are a good source of bioactive peptides. In addition to their high protein content, they contain high levels of dietary fiber, vitamins, and minerals. The aim of this review is to present new bioactive peptides derived from legume seeds and showing inhibitory properties against MS. These compounds may find application in MS diet therapy or functional food production.

## 1. Introduction

Currently, lifestyle, nutrition, stress, and environmental pollution are known to have a key impact on the occurrence and development of many diseases. Especially nowadays when the world has been struggling with the COVID-19 pandemic, such risk factors as poor nutrition, reduced physical activity, stress, and weakening of the body may lead to the development of many diseases, including obesity, diabetes, hypertension, dyslipidemia, or mental diseases that may result in metabolic syndrome (MS).

MS is defined as the presence of risk factors for the development of abdominal obesity, hypertension, high triglycerides, low HDL cholesterol, high blood glucose levels, and insulin resistance. Its prevalence is estimated to be increased in highly developed countries such as the USA, Japan, and European countries, but the biomarkers of MS can vary individually [1]. In addition to the lifestyle, genetic and environmental factors also predispose patients to MS. It is also known that obesity and insulin resistance play an important role in the pathogenesis of MS [2].

Obesity, which is considered a systemic inflammatory condition, has become a global problem especially among residents of highly developed countries. It affects not only adults but also children and adolescents [3]. This multifactorial disease is associated with excessive body weight and adipose tissue deposition caused by consumption of too many calories relative to the demand. The modern way of life, especially that led by young people, associated with low physical activity and sedentary lifestyle promotes overweight and obesity. Obesity increases the risk of non-communicable diseases and leads to disability and death. The formation of excessive body fat is associated with, e.g., high activity of lipases in the organism. These enzymes hydrolyze dietary fats, releasing fatty acids. Excessive activity of these enzymes can lead to fat formation. Therefore, it is advisable to eat foods rich in lipase inhibitors, especially pancreatic lipase, which contributes to the inhibition of the development of obesity [4]. Obesity is associated with excessive calorie intake relative to its consumption by the organism. Long-chain fatty acids are the main source of energy from food. Therefore, reduction of their intake or inhibition of the activity of fat-degrading enzymes, including fatty acids, is one of the most important ways to prevent obesity and obesity-related diseases [5]. Pancreatic lipase is an enzyme hydrolyzing fat in the digestive system. Inhibition of its activity reduces the release of fatty acids and the formation of adipose tissue. Thus, the inhibition of pancreatic lipase can effectively reduce triglyceride absorption, helping to prevent the development of obesity [6]. Currently, there is only one drug on the market that effectively inhibits pancreatic lipase activity, i.e., orlistat. Orlistat can promote weight loss and effectively improve glycemic control in diabetic patients. It has also been shown to affect fat absorption, leading to hyperoxaluria. Therefore, it is necessary to develop natural pancreatic inhibitors [7]. Various protein hydrolysates, peptide fractions, and peptides from various protein sources have exhibited anti-obesity properties, as they can modify dietary lipid metabolism and absorption [8]. The relationship between the specific sequences of peptides and their activity is still unknown; hence, further research is neededOne of the consequences of overweight and obesity can be the development of hypertension. It is associated with the activity of enzymes that make up the renin-angiotensin-aldosterone system. The key enzyme of this system is the angiotensin I-converting enzyme (ACE). It hydrolyzes inactive angiotensin I to angiotensin II, which exhibits vasoconstrictor properties. Excessive activity of this enzyme causes muscular distension and narrowing of the vascular lumen, resulting in an increase in blood pressure. Therefore, one of the therapeutic methods used to treat hypertension is the use of synthetic ACE inhibitors, which very often cause side effects such as cough or rash; hence the search for natural ACE inhibitors derived from food ingredients that may support hypertension pharmacotherapy and diet therapy [9].

Insulin resistance is a pathological condition where the sensitivity of tissues to insulin is reduced, resulting in reduced uptake and utilization of glucose by cells. A low-carbohydrate diet can effectively improve the insulin resistance levels (insulin). In addition, inhibition of enzymes involved in carbohydrate hydrolysis and glucose release such as dipeptidyl peptidase IV, α-glucosidase, and α-amylase, may also have a role in diet therapy and supportive treatment of insulin resistance and prevent the development of diabetes [8,10].

Therefore, new inhibitors of enzymes involved in MS pathogenesis are being sought. These compounds can be derived from food and can support MS pharmacotherapy. Food components showing enzyme inhibitory activity include peptides. The precursors of bioactive peptides are proteins from which these compounds are released mainly through enzymatic digestion or fermentation. They are an important component of functional foods and nutraceuticals in preventing the development and supporting the pharmacotherapy of many diseases. In addition to their inhibitory activity against enzymes, bioactive peptides exhibit antioxidant [11], anti-inflammatory [12], antimicrobial [13], or anticancer properties [14]. Products with high protein content such as legumes, cereals, meat or seafood are good sources of bioactive peptides. New protein sources, e.g., edible insects, are increasingly being used to obtain peptides [15]. Bioactive peptides have also been isolated from low-protein products such as fruits and vegetables and can also be obtained from food by-products [16]. The most common process yielding bioactive peptides is hydrolysis carried out with the use of proteolytic enzymes of plant, animal, or microbial origin. Until recently, it was thought that bioactive peptides were di or tripeptides. It is now known that these compounds can contain more amino acid residues. The exact relationship between the structure and properties of the peptides has not yet been clarified. Food-derived peptides are released in the human gastrointestinal tract during enzymatic hydrolysis; next, they are absorbed and transported to their destinations or can be incorporated into fortified foods or novel food products [17].

Bioactive peptides also have the potential to improve glucose homeostasis and insulin sensitivity by affecting various targets in the body, including carbohydrate digestion, gut hormone release, insulin secretion and action, glucose uptake, and fat tissue modification [10].

Legumes are one of the food ingredients with high protein content. Their protein content varies from 20–35%, depending on the species. In addition, legumes are high in dietary fiber and have a low glycemic index, which is important in MS diet therapy [18].

This review focuses on the characterization of bioactive peptides obtained from legume seeds inhibiting the activity of enzymes involved in MS pathogenesis. The possibilities of obtaining bioactive peptides from legume seeds and their use in MS diet therapies are demonstrated.

## 2. Peptides against Hypertension

Hypertension is defined as permanent high blood pressure over 140/190 mmHg. It is one of the risk factors of cardiovascular disorders. The source of the disease is mainly found in the renin-angiotensin-aldosterone system (RAA), where a glycoprotein named the angiotensin-converting enzyme plays a key role, as significant as that of renin. Disorders in the activity of bradykinin, i.e., a molecule functioning as a vasodilator hormone, are closely connected with ACE effects as well [19].

There are two steps in the primary mechanism: catalysis of angiotensinogen to angiotensin I by rennin and conversion thereof to angiotensin II by ACE, which is manifested in increasing blood pressure. This process is reflected in the activity of aldosterone, which causes an increase in the level of Na^+^ ions and deactivation of the bradykinin hormone [20]. Most medicines available on the pharmaceutical market are based on ACE inhibitory activity, but there are also renin inhibitors. However, Wang et al. [21] questioned their effectiveness in comparison with ACE inhibitors.

Various methods are used for the extraction of peptides from legumes, e.g., solvent extraction, enzymatic hydrolysis, or fermentation [19]. In a study conducted by Yuchen and Jianping [22], thermolysin was applied to obtain peptides from soybean. Additionally, pepsin and trypsin were used in the second and third steps. An ACE inhibition assay was used in each part and the IC_50_ value of the hydrolysates was determined to be 53.5 µg/mL, 51.8 µg/mL, and 115.6 µg/mL, respectively. It was suggested that trypsin was less effective in yielding ACE inhibitory hydrolysates. In total, 12 dipeptides, 10 tripeptides, 7 tetrapeptides, 4 pentapeptides, and 1 hexapeptide were identified. Among them, 1 dipeptide (LW) and 5 tripeptides (LLF, LNF, LEF, LSW, IVF) were highlighted, as they were characterized by a predicted IC_50_ value lower than 10 µM using the QSAR model. The IC_50_ values of all di- and tripeptides are shown in Table 1.

The method of fermentation with *Lactobacillus plantarum* 299 v was used for the release of bioactive peptides from pea seeds; however, no antihypertensive capacity was noted in any sample. Nonetheless, after in vitro digestion and dialysis with a membrane tube, all samples indicated significant IC_50_ values compared to the control sample. The highest inhibitory ability was exhibited by a sample fermented at 22 °C for 7 days (0.19 mg/mL in comparison to the control: 0.37 mg/mL). The separation carried out on Sephadex G10 revealed two fractions with IC_50_ values of 64.04 and 141.27 µg/mL. The first sample was subjected to LC–MS/MS analysis to determine the amino acid sequence—KEDDEEEEQGEEE [23].

Six polypeptides (DALEPDNRIESEGGLIETWNPNNRQ, FEEPQQSEQGEGR, GSRQEEDEDEDE, WMYNDQDIPVINNQLDQMPR, RGEDEDDKEKRHSQKGES, and RLNIGSSSSPDIYNPQAGR) were identified by Jakubczyk et al. [24] in faba bean seeds fermented with the use of *Lactobacillus plantarum* 299 v. The product was hydrolyzed in conditions simulating the digestive system and fractionated to <3.0 kDa peptides. The inhibitory capabilities of the fractions were assessed in different conditions. Nevertheless, the fraction with the lowest IC_50_ value was chosen (30 °C, 3 days, 1.01 mg/mL) and separated into three small parts with IC_50_ values of 0.05 mg/mL, 0.17 mg/mL, and 0.07 mg/mL, respectively. It is reported that the ACE inactivation ability of polypeptides is connected with the presence of such short peptides as GL, DA, GG, EG, IE, ALEP, GR, GE, QG, PQ, GS, MY, PR, IP, WM, YN, KR, KG, QK, EK, KE, RG, RL, IY, IG, AG, and LN according to the BIOPEP database [24].

In another study, bean seeds were fermented in the same conditions as in Jakubczyk et al. [24]. In this work, two 3.5–7 kDa fractions with significant ACE inhibition activity treated with 22 °C for 3 h (IC_50_ = 0.53) and 30 °C for 3 days (0.28 mg/mL) were chosen. The separation of peptides yielded four fractions from each; additionally, ACE inhibition was examined (0.2 μg/mL for III at 22 °C/3 h and 0.36 μg/mL for 4 at 30 °C/3 days). Using LC-MS-MS/MS, the following sequences were identified: INEGSLLLPH and FVVAEQAGNEEGFE in the 3-h treatment at 22 °C and INEGSLLLPH, SGGGGGGVAGAATASR, GSGGGGGGGFGGPRR, GGYQGGGYGG NSGGGYGNRG, GGSGGGGGSSSGRRP, and GDTVTVEFDTFLSR in the 3-day treatment at 30 °C [25].

Moreover, the highest inhibition activity was determined for the 3.5–7 kDa fraction, which suggests that longer peptides strongly influence MS development as well as di- or tripeptides [26].

Combining methods, such as hydrolysis and fermentation, is an efficient way to optimize the acquisition of bioactive peptides from legumes. This approach was applied in studies based on the use of *Lactobacillus rhamnosus* in comparison to Captopril, i.e., a common medicinal product against hypertension. The data showed that the treatment of soy protein isolates (SPI) with the enzyme and fermentation resulted in an increase in inhibition from 60.8 to 88.24% in comparison to the only enzyme treatment (IC_50_ values: 0.592 mg/mL and 0.980 mg/mL; respectively). These values for Captopril were 94.2% and IC_50_ = 0.005 mg/mL. Besides, the LC-ESI-TOF-MS/MS method was applied to investigate the sequences in the bioactive peptides. The main molecules were represented by IAKKLVLP, PDIGGFGC, PPNNNPASPSFSSSS, GPKALPII, and IIRCTGC, with the first one exhibiting the highest inhibition activity (83%). Nevertheless, additional treatment of the fermented SPI with gastrointestinal enzyme hydrolysis did not cause significant changes in the ACE inhibitory capacity [27].

The use of novel peptide-producing methods, such as high pressure atmosphere, is increasingly being observed. The research conducted by Garcia-Mora et al. [28] proved that these special conditions had an influence on the isolation of bioactive peptides from lentil and their ACE-inhibitory activity. Three of the 4 applied enzymes (Protamex, Savinase, Corolase 7089) exhibited a significant increase in the inhibition with the growing pressure (65%, 69%, and 70% respectively). The hydrolyzate generated with the use of Savinase at 300 MPa for 15 min was subjected to analysis of the content of small bioactive peptides. MALDI-TOF peptide mass fingerprinting carried out after ultrafiltration revealed the presence of the following peptides: DLPVLRWLKL, SRSDQDNPFIF, REQIEELRRL, DLAIPVNRPGQLQ, DLAIPVNRPGQLQSF, and others. It was predicted that the ACE-inhibitory capacity was regulated by di- and tripeptides included in larger molecules containing hydrophobic amino acid residues at the C-domain. The IC_50_ value was not evaluated [28].

Besides the use of raw legumes, flours made from these plants can be a good source of biopeptides. Jakubczyk and Baraniak [29] obtained 3 sequences from lentil flour via an in vitro process in conditions simulating the digestive system and a simulated absorption process. Ion exchange chromatography was employed to investigate the highest content of bioactive peptides, which were subjected to gel-filtration chromatography. Fraction 5 (identified sequences: KLRT, TLHGMV, VNRLM) with an IC_50_ value of 0.0197 mg/mL was isolated and purified. Six fractions were extracted, but the highest ACE-inhibitory capacity was 0.13 mg/mL. It was suggested that lower values were associated with the synergistic action of various peptides against the ACE activity, which was stronger than that of individual molecules [29].

Many in vitro studies indicate the antihypertensive properties of peptides derived from legumes, but there are still few clinical trials. A study conducted by Aluko et al. [30] involved isolation and analysis of the efficacy of antihypertensive peptides derived from pea proteins. Pea protein isolate was hydrolyzed using thermolysin, and peptide fractions were obtained using a 3-kDa ultrafiltration membrane. Peptides with the following sequences were isolated from the fraction with the highest properties: LTFPG, IIPLEN, LSSGDVF, IFENLQN, and FEGTVFENG. LTFPG, IFENLQN, and FEGTVFENG were characterized by higher inhibitory properties against renin and ACE; therefore, they were administered orally to spontaneously hypertensive rats at a dose of 30 mg/kg body weight. The LTFPG administration caused the fastest decrease in systolic blood pressure with a maximum of −37 mmHg after 2 h. In contrast, the maximum effects of IFENLQN (−37 mmHg) and FEGTVFENG (−25 mmHg) were observed after 4 h. The results of the study indicate that pea protein can be used as an ingredient in functional and dietary foods.

In addition, mung bean proteins also serve as precursors of peptides with blood pressure-lowering properties. Hydrolyzed proteins obtained using bromelain yielded ACE- and rennin-inhibiting peptides with the following sequences: LPRL, YADLVE, LRLESF, HLNVVHEN, and PGSGCAGTD. The results showed that LRLESF was the most potent ACE inhibitor (IC_50_ = 5.4 μM), while LPRL had the weakest effect (IC_50_ = 1912 μM). YADLVE inhibited renin activity at a level of 97%, while LRLESF caused only approximately 30% inhibition. These peptides lowered blood pressure in spontaneously hypertensive rats by 36 mmHg. It is noteworthy that YADLVE exerted a sustained effect 24 h after administration, indicating that it can be used as a blood pressure-lowering product. However, further studies are recommended [31].

As demonstrated by the research, bioactive peptides contained in legumes can inhibit the angiotensin-converting enzyme with unquestionable efficiency. However, the method for extraction thereof is an important factor in the final outcomes, e.g., peptide sequences.

## 3. Peptides against Obesity and Insulin Resistance

Obesity is a multifactorial disease that impairs the functioning of the entire body and causes disorders of the circulatory system. It is characterized by excessive accumulation of body fat due to an imbalance between energy intake and energy expenditure caused by lifestyle and improper eating habits. Therefore, changing the key factors contributing to the development of obesity is an important aspect in the treatment. Increased physical activity and a proper diet reduce the effects of obesity. Unfortunately, this often long-term approach and its recommendations are not followed by patients; therefore, it is necessary to use pharmaceutical products inhibiting the digestion of nutrients, especially fats, which are the main source of unwanted calories in the diet. Pancreatic lipase is one of the enzymes that cause the release of fatty acids from fatty foods. Drugs used to treat obesity are based on inhibition of this enzyme. Despite the documented positive effects in combating the incidence of obesity, the drugs also have side effects mainly in the gastrointestinal tract, e.g., bloating, diarrhea, or abnormal liver function. Therefore, researchers are looking for alternative compounds with pancreatic lipase inhibitory effects that have no side effects and occur naturally in foods. One such alternative could be peptides occurring naturally in food or released from food under the influence of enzymes in the digestive tract. The exact relationship between the structure of peptide pancreatic lipase inhibitors derived from legume proteins and their activity is not well understood. A study conducted by Ngoh et al. [32] showed that peptides from Pinto beans inhibited lipase activity in the range from 23 to 87%. This study highlighted the effectiveness and preventive action mechanism of peptides not only in obesity but also in hyperlipidemia and hypercholesterolemia. The results also showed that hydrophobic amino acid residues (i.e., Ala, Leu, Pro, Phe, Gly, Met, and Trp) in peptides have a beneficial effect on inhibition of pancreatic lipase. Therefore, it has been proposed that the presence of hydrophobic residues should be a characteristic feature of lipase inhibitors.

α-Amylase is an enzyme participating in digestion of polysaccharides and appearing in saliva and the duodenum. Briefly, the mechanism is based on breaking down alpha-1,4-glycosidic links, which connect D-glucose units, where glucose and maltose are the final products. A high and increasing concentration of simple sugars in the blood is reflected in hyperglycemia associated with insulin resistance and diabetes [33].

In a study on the sequences of pinto bean bioactive peptides (PBBPs) conducted by Ying-Yuan et al. [33], phage display was used. The method is based on evaluation of protein–protein, protein–peptide, and protein–DNA connections with the application of bacteriophages to combine the protein with the genetic information which encodes it. The use of the phage ELISA test indicated that 5 of the 11 phage-cloned peptides had the highest binding interaction with α-amylase. They were subjected to chemical peptide synthesis (SyP1, SyP3, SyP6, SyP7, and SyP9). SyP9 and SyP1 exhibited the lowest IC_50_ values of 1.97 mg/mL and 8.96 mg/mL, respectively. The peptides were characterized by sequences LSSLEMGSLGALFVCM and PPHMLP [33].

The same peptides were investigated using AR42J, a pancreatic cell line with amylase-containing granules. The ability of PBPs (Pinto bean peptides) to inhibit α-amylase and the viability of cells exposed to the peptides were assessed; additionally, a kinetic assay and identification of the inhibition were conducted. According to the data, PB9 and PB7 were characterized by the highest enzyme inhibitory activity (IC_50_ = 0.3 mM, IC_50_ = 5.92 mM, respectively). It was shown that the cell viability decreased as the concentration of PBPs increased, but the concentration was not toxic even with high doses of PBPs (30.6–99.0% for PBP1, 30.5–83.2% for PBP3, 26.6–82.9% for PBP6, 21.8–73.3% for PBP7, and 47.0–69.0% for PBP9). Uncompetitive and unusual α-amylase inhibition modes were demonstrated [34].

Moreover, it should be noted that the dipeptidyl peptidase-IV (DPP-IV) enzyme has a strong influence on glucose balance in blood. Briefly, DPP-IV decreases the concentration of incretins, such as GLP-1 (glucagon-like peptide-1), which causes glucagon release, increases the blood sugar level, and inhibits insulin release. Hence, DPP-IV inhibitors may be regarded as good therapeutic targets for patients [35].

A study carried out by Mojica and González de Mejía [36] demonstrated the protein profiles of 15 common bean cultivars. After hydrolysis with pepsin or pancreatin, two sequences of the most predominant peptides (among 74) with DPP-IV inhibition capacity: LLAH and YVAT were obtained. Among all peptides, α-amylase inhibitors have been investigated, but the highest percentage of inhibition was equal 14.9% (compared to the positive control, i.e., acarbose (AC) characterized by 100% inhibition) [36]. Legumin type B and vicillin from faba flour were found to be other popular sources of bioactive peptides with DPP-IV inhibition activity. In silico digestion showed sequences MSKP, FL, TSTC, ATSSE, NQCR, etc, in legumin and MAATT, DSF, GIA, ASVC, ESNR, etc. in vicillin. These two types of protein are considered a good source of low molecular weight peptides inhibiting various enzymes [37].

An atypical source of α-amylase inhibitors is *Mucuna pruriens* from the Fabaceae family. This legume is abundant in proteins, which were obtained in the study by applying extraction and purification. Certain techniques, e.g., ammonium sulfate fractionation, ion exchange, and gel filtration chromatography, were used for isolation and purification of peptides with inhibition capability. The molecules were characterized by inhibitory activity expressed as an α-amylase inhibitory unit, i.e., the amount of α-amylase inhibited in the evaluation conditions. The IC_50_ value and sequences of the protein were not evaluated [38].

α-Amylase inhibitors were investigated in a study of two chickpea cultivars, labeled as W4 and W2. Human saliva, α-amylases from porcine pancreas, maize, and *Bacillus subtilis* were studied. In the research, ammonium sulfate precipitation (ASP) was used with three fractions of 0–60%, 60–80%, and 80–100% of each cultivar. Two fractions (60–80% and 80–100%) were mixed and subjected to ion exchange chromatography (IEC). Among seven fractions from W2 and eight from W4, two fractions with the highest inhibitory activity, 54.55% and 65.24%, respectively, were chosen. The next step showed fraction PIII from W2 as the strongest α-amylase inhibitor (80.85%) during the reversed phase liquid chromatography (RPLC) process. SDS-PAGE indicated one band with a molecular mass about 25 kDa. The same methodology was applied to W4, where the highest inhibitory activity, i.e., 75.93%, was exhibited by fraction PIII. Interestingly, the same proteinaceous particle as that in the W2 cultivar was discovered. The UPLC/MS-MS results revealed the presence of 3 polypeptides: GKEVYLFK, FCALIDYAPHSNK, and FCALIDYAPHSNKDK, which were compatible with chickpea lectin CAL found in the GenBank. It should be noted that CAL is not part of the lectin-arcelin-aAI1 supergene family. The IC_50_ value was not indicated [39].

In addition to α-amylase and DPP-IV enzymes, α-glucosidase is regarded as another diabetic-dependent enzyme. Breaking down bonds between molecules building oligosaccharides is its essential function leading to an increase in the level of reducing simple sugars [40]. An inhibition assay of these enzymes was carried out based on two hard-to-cook common bean varieties: Negro 8025 and cv. Pinto Durango. The products were subjected to alcalase and bromelain hydrolysis separately; subsequently, they were both subjected to a pepsin-pancreatin treatment. After gel electrophoresis (SDS-PAGE), sequences were identified with the use of HPLC–ESI–MS/MS. The LLSL and DFFLS sequences distinguished among the peptides were found to be DPP-IV inhibitors derived from both cultivars. The highest percentage of inhibition of each enzyme was exhibited by <1 kDa fractions. The strongest influence on α-glucosidase was shown by pinto Durango beans hydrolyzed with alcalase (76.4%), but it was not significantly higher than that of Negro 8025 beans assessed using the same method. The α-amylase inhibitory capacity was assessed for pinto Durango beans hydrolyzed with bromelain (~50%). Fractions <1 kDa were found to exert various but not significantly different effects on dipeptidyl peptidase IV activity (about 55%), as shown by the different hydrolysis methods applied to the different types of product. The IC_50_ value was not calculated [41].

An untypical method for acquisition of bioactive peptides is encapsulation applied by Cian et al. [42] with the use of maltodextrin/gum Arabic. Lima beans were used as a protein source and subjected to hydrolysis by Alcalase^®^ and Flavourzyme^®^ and the protein release (at 2.0 and 7.0 pH) process, which yielded samples labeled as pH-MC. Additionally, the products were subjected to simulated gastrointestinal digestion and marked as GID-H (hydrolysates after digestion) and GID-MC (microcapsules after digestion). All samples were compared in terms of their inhibition properties against α-glucosidase, α-amylase, and DPP-IV activity. The highest IC_50_ values were noticed for GID-H in all tests, which is equal to the lowest inhibitory activity of the peptides. Samples after hydrolysis, the protein release process, and the microcapsule digestion process did not show significant differences. Nevertheless, encapsulating is associated with prevention of bioactive properties against gastrointestinal conditions. No sequences were identified in the study [42].

Legumes can be an enrichment factor in food products as well. The impact of fermented bean seed flour (BF) addition to wheat wafers on α-glucosidase activity has been assessed in another study. To this end, the researchers used different concentrations of the supplement (10–50%), 100% BF wafers, and 100% wheat flour wafers. The results showed the lowest IC_50_ = 0.14 mg/mL in products with the 40%, 50%, and 100% content of BF. The sequences were not investigated [43].

The aim of the study conducted by Mojica et al. [44] was to evaluate in silico, in vitro, and in vivo effects of a black bean protein hydrolysate and pure peptides AKSPLF, ATNPLF, FEELN, or LSVSVL on glucose absorption. The protein hydrolysate was prepared with the use of Alcalase. The results indicated that the peptides blocked glucose transporters GLUT2 and SGLT1. Studies on Caco-2 cells indicated that the hydrolysate applied at a concentration of 10 mg/mL reduced the formation of intracellular reactive oxygen species by 71% and reduced glucose absorption by 21.5% after 24 h. Oral glucose tolerance tests in rats showed a 24.5% decrease after the glucose meal (50 mg hydrolysate/kg bw). In a rat model of hyperglycemia, the hydrolysate decreased blood glucose in a dose-dependent manner. The lowest fasting glucose levels were found in rats receiving 150 and 200 mg/kg BW/day of the hydrolysate. The hydrolysate obtained from black beans is an inexpensive dietary source of bioactive compounds that can be used to control blood glucose levels.

Glucose and insulin metabolism disorders leading to obesity can be affected by bioactive peptides from pulses. Many different methods are used to extract these molecules (Table 2). However, there is no research describing sequences characterized by inhibition activity.

## 4. Anti-Inflammatory and Antioxidant Peptides as Prevention Agents against MS

The inflammatory process can be divided into acute and chronic. The former is a physiological reaction to injuries or infections and is related to normal immune response. The latter is associated with various pathological changes. Acute inflammation usually lasts few days, which is the difference between these two types of reaction. It is proven that chronic inflammation is one of the most significant components of the development of such diseases as cardiovascular, autoimmunological, and metabolic diseases as well as hormone imbalance [45].

Oseguera-Toledo et al. [41] evaluated the anti-inflammation capacity of Pinto Durango and Negro 8025 beans treated with alcalase and bromelain. The activity was determined with the use of oxygen radical absorbance capacity (ORAC) and NO radical scavenging capacity assays. Among all fractions in the mass ranges <1 kDa, 1–3 kDa, 3–5 kDa, 5–10 kDa, and >10 kDa, the 5–10 kDa group had the highest ORAC value (708.0–932.6 mmol TE/g) in each group. In another study, beans were subjected to alcalase–flavourzyme and pepsin–pancreatin hydrolysis, which revealed stronger anti-inflammation activity with a value of 8.1 mM/mg [46]. In turn, the 1–3 kDa group of fractions exerted the highest impact on nitric oxide inhibition (56.4–68.29%). The authors identified a few bioactive peptide sequences, e.g., YAAHEV from Pinto Durango beans and NEGEAH from both Pinto Durango and Negro 8025 beans [41].

The FDPAL peptide was discovered in a study of in vitro and in vivo antioxidative capacities of a soy protein isolate. The material was subjected to hydrolysis with alcalase and ion-exchange chromatography, which revealed 3 fractions with the strongest antioxidant capacity. Fraction II was separated by Sephadex G10. Further, fraction I was subjected to LC-MS/MS, which facilitated the identification of sequences. To corroborate the anticipated functions, Fenton’s reaction and a pyrogallol self-oxidation assay were carried out with vitamin C as a control group. Both tests revealed a considerable free radical scavenging effect. Additionally, an MTT assay was conducted to determine the preventive effect of FDPAL on a HeLa cell culture. The bioactive peptide with a molecular mass up to 10 mM was correlated with rising of viability with H_2_O_2_ presence [47]. The in vivo part was carried out on a worm species *Caenorhabditis elegans* to establish the longevity-stimulating effect of FDPAL. The experiment indicated a significant protective effect of the bioactive peptide against a pro-oxidant compound—juglone, which promotes oxidative stress in tissues by producing superoxide anions. The data showed a 24.2% increase in the survival rate in the treatment with the peptide, compared to the control group [47].

Faba bean has proven health-promoting capacities, e.g., antioxidant effects confirmed by tests of its protein extracts and hydrolysates obtained with the use of various enzymes. Their ABTS^·+^ and DPPH^·^-radical scavenging activities and iron chelating ability were assessed. Alcalase- and trypsin + pepsin-treated samples exhibited the highest antioxidative activity against ABTS, i.e., approx. 55 and 60 AA eq/g of protein. Regarding DPPH, the strongest influence on the radical was indicated by the alcalase- (about 24 AA eq/g of protein) and pepsin + trypsin-treated samples (approx. 26 AA eq/g of protein). Moreover, the alcalase-hydrolysis was more effective than the mixture of the other enzymes. Besides, the protein extract was associated with considerably lower antioxidant properties in the iron chelating assay. The most relevant results were exhibited by hydrolysates, but those obtained with the use of pepsin, trypsin, and trypsin-alcalase (trypsin—8.62 and trypsin-alcalase—7.86 mg/L) were characterized by lower Fe^2+^ chelating properties. The researchers identified some bioactive peptides in the faba bean hydrolysates that share sequences with earlier reported antioxidant peptides, such as TETWNPNHPEL, ALEPDHR, VIPAGYP, PHW, PHY, YVE, and others [17].

A mice-model experiment showed a significant impact of pea peptides on the immune system in a study focused on the problem of fatigue. During tests, some basic components of chronic inflammation, such as interleukins-2, -4, and -6, IFN-c, and TNF-a, were evaluated. Additionally, phagocytosis and sIgA were assessed. Five groups were designed: a control group (CG), a group fed with Isolated Pea Protein (IPP), and groups receiving low (PPL), medium (PPM), and high (PPH) amounts of Pea Peptides. Pea protein was subjected to hydrolysis with compound protease (90% alkaline protease and 10% papain). The statistical data showed high significance of the phagocytosis effect, compared to CG and IPP. Considering the next factor of the first immune defense line, the results of sIgA exhibited the same trend. Among cytokines, the TNF-α in PPH group was characterized by the lowest value (8.47 ng/mL), which indicated the strongest anti-inflammatory activity of the peptides. It should be emphasized that the level of IL-6 in the PPL, PPM, and PPH groups was significantly different from that in the other groups. The MALDI–TOF–MS analysis facilitated extraction of 3 peptides: QLEELSK, KGDFELVGQ, and FFELTPEKNQ [48].

In a study on soy flours, their antioxidant functions were assessed using the Oxygen radical absorbance capacity (ORAC) assay involving reduction of radicals. First, germinated (for 18 and 72 h; G18 and G72) and non-germinated (G0) flours were subjected to alcalase hydrolysis (for 1, 2, and 3 h). As shown by the data, the 1-hour hydrolysis process increased the antioxidant capacities. Besides, the germination process contributed to enhancement of these abilities in samples hydrolyzed for the same time as the non-germinated flours. The highest value was noted in the 1-h hydrolysis variant (740.6 mM TE/g). Cyclooxygenases-1 and -2 (COX-1 and COX-2) were used as other inflammatory markers. The inhibitory activity of the extract and hydrolysates was measured by determining the amount of PGF2_α_. Emphasizing the stronger impact of COX-2 on inflammation, the highest inhibition capability was exhibited by the non-hydrolyzed samples germinated for 18 and 72 h [49].

Other well-known factors playing a significant role in the development of chronic diseases, such as nitric oxide (NO), prostaglandin E_2_ (PGE_2_), and tumor necrosis factor α (TNF-α), were tested. Plated murine macrophage cell line RAW 264.7 was used to evaluate the release of inflammation triggers. A significant effect of hydrolysis on reduction of NO release was observed in all samples. On the other hand, the longer germination process was associated with a weaker impact on NO production. However, considering the hsame time of lysis, samples subjected to the 18-h germination exhibited the lowest value of produced TNF-α. Additionally, the non-germinated flour was characterized by lower reduction of TNF-α release. Moreover, the highest values of PGE_2_ were associated with the 72-h germination process, and the highest reduction was shown by the G0 sample subjected to 2-h hydrolysis [49].

The research presented above has evidenced the anti-inflammatory activity assessed with a wide range of methods (Table 3). Bioactive peptides from legumes have an influence on the COX-1 and COX-2 enzymes and inhibit the production of cytokines. Such molecules as interleukins or TNF-α are regarded as promoters of chronic inflammation in the pathogenesis of various diseases.

## 5. Diet Factors vs. Metabolic Syndrome

Diet patterns have been proved to exert an impact on the development of MS components such as insulin resistance, dyslipidemia, obesity, and hypertension. The number of consumers of vegan and vegetarian diets has been observed to increase, and the Mediterranean Diet (MD) is considered a healthy choice as well. Between 2014 and 2017, there was a 5-fold increase in the number of plant-diet followers [50]. Various studies show differences between vegan and non-vegan diet patterns. A comparative study of the BMI index and T2DM incidence in consumers and non-consumers of meat and animal-derived products demonstrated a decrease in these two parameters in favor of the vegan diet [51].

A healthy diet is based on whole grain cereals, flours, and groats as well as nuts, fresh vegetables and fruits, legumes, unsaturated oils, and fermented food. Additionally, good quality cheeses, fish, and meat in reasonable amounts are acknowledged as healthy components of MD. Spices and herbs rich in multiple bioactive substances play a relevant role in the proper nutrition model.

A well-sustained diet includes fiber, i.e., a component of fruits and vegetables that influences the gut microbiota condition and improves its immunomodulatory activity. Besides, fiber-containing food contributes to the growth of Actinobacteria and Bacteroidetes, but inhibits the growth of Firmicutes associated with obesity [52]. Clinical trial results showed that greater abundance of Firmicutes was connected with higher absorption of calories compared to a Bacteroidetes-rich microbiome [53]. Various bacteria colonizing intestines play an essential role in sustenance of health. This was proved in a study on mice fed a fat-rich diet and receiving the *Lactobacillus plantarum* K12 probiotic strain. The supplemented mice exhibited significant weight loss in comparison with the non-supplemented mice. The former group was characterized not only by lower energy absorption but also by lower content of leptin, whose high level is typical in obese patients, and lower amounts of triglycerides in blood [52]. Additionally, the colon-colonizing microbiota is responsible for fermentation of previously undigested fiber, which leads to the production of short-chain fatty acids (SCFAs) such as butyrate, propionate, and acetate. SCFAs have a massive impact on gut microbiome quality and well-being. For instance, they have a trophic effect, which enhances the process of intestinal epithelium regeneration. Butyric acid is the main energy source for colonocytes and exhibits anticancer activity through activating the apoptosis process and inhibiting proliferation in neoplastic cells [54]. Forty three young men classified in a randomized study were divided into two groups: consuming high-protein legume-based meals and high-protein animal-based meals. The participants from the plant group were characterized by higher satiety levels after consumption of ~95 less calories than the animal group. SCFAs are related to stimulating the release of peptide YY (PYY) and glucagon-like peptide (GLP-1) [55]. A positive influence of high wheat fiber consumption was observed in patients with hyperinsulinemia, while SCFAs and GLP-1 increased, compared to patients eating low amounts of wheat fiber. Therefore, bacteria present in human intestines are involved in appetite regulation [56].

Low calorie density, typical for Mediterranean and plant-based diets, has an impact on satiety. This is associated with the feeling of hunger and satiety, which are strictly related to obesity. Neurotransmitters in the hypothalamus, such as the neuropeptide Y/agouti-related protein (NPY/AgRP) and proopiomelanocortin (POMC) are involved in appetite control by influencing glucose, insulin, ghrelin, leptin, and peptide YY [57]. In a study on a modified rat diet, inclusion of different types of fiber caused weight loss and lower food intake by 10%, 17%, and 19%, depending on the fiber type [58]. When the calorie density is lower, a bigger portion of meal can be consumed and provide a feeling of fullness. This also applies to the differences between the intake of whole fruits and the consumption of juices, which are not rich in fiber. The results of the study showed advantages of consumption of three apples or three pears versus 3 oat cookies per day for 10 weeks. In the groups receiving the fruit-based meals, reduced calorie intake and significant weight loss were observed [59]. The food volume is regarded as a factor stimulating the stomach to stop ghrelin production and leptin release from adipocytes. Leptin resistance is a pathology mechanism in patients with obesity or overweight. A study conducted on obese mice indicated that sulforaphane, occurring in cruciferous vegetables, had a relevant impact on intraperitoneally administered leptin and improved sensitivity to the hormone. As shown by the data, single and triple injections suppressed food consumption and body weight gain [60]. One of the methods for losing weight is to reduce the amount or volume of consumed meals. Persistent hunger may be a result of these actions. For this reason, an increase in satiety through reduction of the caloric density of the product rather than its amount may be considered. In a crossover trial, 24 normal-weight young women received an ad libitum diet with a 25% smaller volume of meals. This resulted in a 10% reduction of calorie consumption. At a level of 25% energy reduction in the food portion supplied (with the same weight of food), the calorie intake declined by 24% [61]. Controlling food absorption is an adequate means of prevention of obesity development.

High carbohydrate and fat consumption in believed to be the trigger of insulin resistance and type 2 diabetes (T2D) development. However, glucose imbalance is caused not only by excessive carbohydrate intake but mostly by a poorly balanced diet containing processed food, refined sugar, and oils and, not surprisingly, by lack of physical activity. Whole grain products and good quality oils can have a positive influence on glucose and insulin blood levels and insulin sensitivity [62]. Glycated hemoglobin (HbA1c) is an indicator of glycemic status in addition to standard sugar and hormone measurements. HbA1c is a complex of hemoglobin and glucose characterized by 120-day durability; hence, it is a reliable factor for long-term evaluation of diabetes [63]. Esposito et al. [64] performed a meta-analysis of trials examining the correlation between MD and T2D. Three intervention studies were set up to compare two different MDs and the same control diet. The MDs were associated with a lower glycated hemoglobin amount corresponding with the blood sugar level. The same results were obtained in four different analyses where the HbA1c range dropped from −0.3% to −0.47% [64]. Fat-content is regarded as a relevant component of the anti-diabetes diet pattern. Saturated fats present in processed food and derived from animals are related to disease development. Briefly, they are associated with lipotoxicity leading to β-cell apoptosis induced by nitric oxide (NO). In turn, unsaturated fatty acids have been proved to enhance insulin sensitivity. Moreover, eicosapentaenoic acid (EPA) and docosahexaenoic acid (DHA) are tightly related to lower fasting blood glucose. Additionally, α-linolenic acid (ALA) influences plasma GLP-1, which intensifies glucose-dependent insulin secretion [65]. This was evidence in a cross-sectional study of 3383 Japanese subjects, where insulin-resistance was reduced in normal-weight patients [66]. Likewise, ALA-rich flaxseed intake reduced HOMA-IR index by 23.7% [67]. Diabetes and glucose management in the body can also be sensitive to the effects of herbs and spices. One of the ingredients with the most widely investigated activity is cinnamon showing anti-inflammatory, anti-microbial, and anti-cancer properties. A triple-blind randomized clinical trial was focused on investigation of the activity of cinnamon in patients with diabetes mellitus type II. Interestingly, the researchers observed improvement in various indicators between the test group and the control group. Patients consuming cinnamon exhibited a significant decrease in fasting plasma glucose, two-hour postprandial blood glucose, HbA1c, insulin, and HOMA-IR. Additionally, the triglyceride content, total cholesterol, and HDL and LDL cholesterol levels were improved [68].

Therefore, plant-based diets seem to have a more beneficial effect on human health. Besides the inclusion of fruits, vegetables, and nuts, the processing level should be taken into consideration. In a cross-sectional PREDIMED-Plus study, three patterns of plant-based diet have been investigated. The aim of the research was to evaluate the association between cardiometabolic risk in adults and nutrition. General pro-vegetarian (gPVG), healthy pro-vegetarian (hPVG) and unhealthy pro-vegetarian (uPVG) food patterns were assessed. Within the unhealthy and healthy PVG diets, some kinds of food were separated, e.g., potatoes were divided into fried/chips and cooked/roasted groups, and grains were divided into whole and refined cereals. In total, 6439 overweight or obese participants meeting at least 3 criteria of metabolic syndrome were included in the study. As shown by the results, gPVG and hPVG were related with lower cardiometabolic risk compared to uPVG [69].

In a study evaluating the relationship between the intake of legumes and nuts and metabolic syndrome, 420 nurses took part in the clinical trial. The participants were divided into quartiles, depending on portion consumption. A food frequency questionnaire (FFQ) was conducted. However, the outcomes did not prove any significant relationship between the intake of particular legumes or nuts and MS. Although other studies have indicated this association, it is considered that the described research was more focused on diet patterns than on specific food groups [70].

There are also studies on the role of consumption of specific legumes in prevention of the development of MS diseases. A study has been conducted on the effects of legumes in preventing type II diabetes and certain cardiometabolic diseases, e.g., high blood pressure, obesity, and stroke. An exploratory study was conducted in a population (n = 468) from Diamare division (Far North, Cameroon). The results generally showed that the consumption of soybeans, Bambara beans, and cowpea was associated with the prevention of metabolic diseases in the study population. The results were influenced by the content of active ingredients in the seeds, such as proteins, phenolic compounds, and dietary fiber. Since legume seeds are a good source of peptides and can be used in the production of hydrolysates, they can certainly be a good resource in the treatment and/or prevention of metabolic and neurodegenerative diseases [71].

Overall, the correlation between diets and chronic inflammatory diseases is undeniable. It is difficult to assess the influence of one group of products on development of disorders, disregarding the whole diet and the variety of diets. Nevertheless, in vitro investigations show a potentially positive impact on MS-related biochemical components and processes.

## 6. Conclusions

There is extensive research on bioactive peptides derived from legume seed proteins, especially given the development of food technology, knowledge of the relationship between the development of diseases and nutrition, and the relationship between the structure of the compound and its activity. It should be noted that further studies are needed to determine the mechanism of action of bioactive peptides on the activity of enzymes involved in the pathogenesis of MS. These studies are complex due to the variety of the structures, charge, mass, and bioavailability of peptides. In addition, orally administered peptides can be hydrolyzed in the gastrointestinal tract, which is another area of research. Legume seed proteins are a good source of bioactive peptides and can form the basis for the development of new functional food products for MS prevention and diet therapy. Currently, there is a lot of research on the effects of a plant-based diet on human health. Determination of the exact functions of dietary components requires clinical studies, which are still limited. It is necessary to combine in vitro and in vivo studies in order to determine the effects of bioactive peptides on human health as much as possible.

## Figures and Tables

**Table 1 foods-11-03300-t001:** IC_50_ values and bioactive peptide sequences of investigated legumes against ACE activity.

Source	Type of Legume	Method	Molecule Indicating IC_50_ Value	IC_50_ Value	Sequences
[14]	Soy	Hydrolysis	Di- and tripeptides	3.4 µM 4.4 µM 4.6 µM 5.2 µM 5.4 µM 6.7 µM	LSW LW LEF LNF IVF LLF
[15]	Pea	Fermentation + digestion	Fraction	64.04 µg/ml	KEDDEEEEQGEEE
[16]	Faba bean	Fermentation + digestion	Fraction	1.01 mg/mL	DALEPDNRIESEGGLIETWNPNNRQ, FEEPQQSEQGEGR, GSRQEEDEDEDE, WMYNDQDIPVINNQLDQMP, RGEDEDDKEKRHSQKGES, RLNIGSSSSPDIYNPQAGR
[17]	Common bean	Fermentation	Fraction	0.2 µg/mL	INEGSLLLPH and FVVAEQAGNEEGFE
				0.36 µg/ml	NEGSLLLPH, SGGGGGGVAGAATASR, GSGGGGGGGFGGPRR, GGYQGGGYGG NSGGGYGNRG, GGSGGGGGSSSGRRP, GDTVTVEFDTFLSR
[19]	Soy isolated protein	Fermentation + proteolysis	U/D	U/D	IAKKLVLP, PDIGGFGC, PPNNNPASPSFSSSS, GPKALPII IIRCTGC
[20]	Lentil	Hydrolysis with high pressure atmosphere	U/D	U/D	DLPVLRWLKL, SRSDQDNPFIF, REQIEELRRL, DLAIPVNRPGQLQ, DLAIPVNRPGQLQSF and more
[21]	Lentil flour	Digestion	Fraction	0.0197 mg/mL	KLRT, TLHGMV, VNRLM

U/D—undefined.

**Table 2 foods-11-03300-t002:** IC_50_ values and bioactive peptide sequences of investigated legumes with activity against enzymes breaking down carbohydrates.

Source	Type of Legume	Enzyme	Method	Molecule Indicating IC_50_ Value	IC_50_ Value	Sequences
[22]	Pinto bean	α-amylase	Synthesis	Sequence	1.97 mg/mL	PPHMLP
[23]	Pinto bean	α-amylase	Synthesis	Peptide	0.3 mM	U/D
[25]	Common bean	DPP-IV	Hydrolysis	U/D	U/D	LLAH, YVAT
[26]	Faba bean flour	DPP-IV	In silico digestion	U/D	U/D	MSKP, FL, TSTC, ATSSE, NQCR, MAATT, DSF, GIA, ASVC, ESNR and more
[27]	*Mucuna pruriens*	α-amylase	Extraction + purification	U/D	U/D	U/D
[28]	Chickpea	α-amylase	Extraction + purification	U/D	U/D	GKEVYLFK, FCALIDYAPHSNK, FCALIDYAPHSNKDK
[30]	Common bean	α-glucosidase	Hydrolysis	U/D	U/D	LLSL, DFFLS
[31]	Lima bean	DPP-IV α-amylase α-glucosidase	Hydrolysis	Hydrolysates	~1.5 mg/mL ~240 µg/mL ~260 µg/mL	U/D
[32]	Bean seed flour	α-amylase	Fermentation + hydrolysis	Hydrolysates	0.14 mg/mL	U/D

U/D—undefined.

**Table 3 foods-11-03300-t003:** Various indicator values and bioactive peptide sequences of investigated legumes with anti-inflammatory capacities.

Source	Type of Legume	Method	Evaluation Method	Molecules with Anti-Inflammatory Activity	Indicator Value	Sequences
[30]	Common bean	Hydrolysis	ORAC	Fraction	708.0–932.6 mmol/TEg	YAAHEV NEGEAH
[34]	Common bean	Hydrolysis	TEAC	Hydrolyzate	8.1 mM/mg	U/D
[35]	Soy protein isolate	Hydrolysis	Pyrogallol autoxidation assay	Sequence	1 mM—50% inhibition	FDPAL
[35]	Soy protein isolate	Hydrolysis	In vivo	Sequence	24.2% of survival rate	FDPAL
[36]	Faba bean	Hydrolysis	ABTS + DPPH	Hydrolyzate	59.7 AAeq/g 28.2 AAeq/g	TETWNPNHPEL, ALEPDHR, VIPAGYP, PHW, PHY, YVE and more
[37]	Pea	Hydrolysis	In vivo (TNF-α secretion)	Hydrolyzate	8.47 ng/mL	QLEELSK, KGDFELVGQ, FFELTPEKNQ
[38]	Soy flour	Germination + hydrolysis	ORAC	Hydrolyzate	740.6 mM TE/g	U/D
[38]	Soy flour	Germination + hydrolysis	COX-2	Extract	~75% inhibition	U/D
[38]	Soy flour	Germination + hydrolysis	PGE_2_	Hydrolyzates	962.8 pg/mL	U/D
			NO		35.3 µM nitrite	
			TNF-α		17.3 pg/mL	

U/D—undefined.

## Data Availability

All relevant data are included in the article.

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
