# Peer review of "Bioactive Peptides Obtained from Legume Seeds as New Compounds in Metabolic Syndrome Prevention and Diet Therapy"

_foods, 2022, doi:10.3390/foods11203300_

Round 1

Reviewer 1 Report

The submitted manuscript is well-designed and written, with a good command of English. Thematically the work is interesting for the researchers and professionals and the proposed manuscript is relevant to the scope of the journal.

The abstract is concise and clearly written, with a clear representation of the aim of the paper. It contains 175 words; therefore it meets the demands of the journal Foods (200 words max).

The title of the paper adequately reflects the subject under investigation in the proposed study.

The literature review is comprehensive and properly done. References are numbered in order of appearance in the text, as demanded by the formatting rules of the journal. However, references are not formatted after journal rules. Although there is no limitation in the number of references, a reference list of 59 citations is completely sufficient to cover the topic proposed.

Considering all,

I suggest this paper to be published in Foods after minor technical revision (comments can be found in PDF file).

Author Response

Thank you very much for all your valuable comments. The publication has been revised according to the comments posted in the .pdf file point by point. In addition to the references section. We are confused because References were prepared using Mendeley program which prepares citations according to the requirements of the journal.

Reviewer 2 Report

The present study focused on the effect of bioactive peptides obtained from legume seeds on the prevention of metabolic syndrome, and these compounds may contribute to metabolic syndrome or functional food production. Although the author comprehensively reviewed the antihypertensive peptide, hypoglycemic peptide and anti-inflammatory peptide from legume, the introduction of lipid-lowering peptide and weight reducing peptide is less. Obesity and dyslipidemia are also components of metabolic syndrome, and the effect on diabetes is not the same as obesity.  some points should be improved as follows:

1) The introduction of bioactive peptides derived from food in the background is less, and therefore relevant concepts, sources and general progress need to be introduced.

2) Bioactive peptides in lipid-lowering should be introduced in the background.

3) The author introduced the research progress of antihypertensive peptide from these three aspects in detail, but what are the problems at present, such as whether to carry out animal experiment verification, etc. Most IC50 measurements are only conducted through in vitro experiments. Have animal experiments and population studies been carried out? What are the specific results?

4) in the section “Peptides against obesity and insulin resistance”, We haven't seen the research results of peptide related to weight loss.

5) about the reference 30: Antioxidant is not equal to anti-inflammatory, and therefore the author needs to carefully organize this part.

6) in the section “Diet factors vs. metabolic syndrome”, It is necessary to emphasize the possible role of legumes in diet, especially the protein hydrolysates of legumes, and not deviate from the theme of this review.

7) Better add the prospect research in the end of the review.

Author Response

Review 2

The present study focused on the effect of bioactive peptides obtained from legume seeds on the prevention of metabolic syndrome, and these compounds may contribute to metabolic syndrome or functional food production. Although the author comprehensively reviewed the antihypertensive peptide, hypoglycemic peptide and anti-inflammatory peptide from legume, the introduction of lipid-lowering peptide and weight reducing peptide is less. Obesity and dyslipidemia are also components of metabolic syndrome, and the effect on diabetes is not the same as obesity. some points should be improved as follows:

We are very grateful for your detailed review of paper. All suggestions were taken into consideration and the manuscript was corrected point by point.

1) The introduction of bioactive peptides derived from food in the background is less, and therefore relevant concepts, sources and general progress need to be introduced.

The introduction section was rewritten.

2) Bioactive peptides in lipid-lowering should be introduced in the background.

The information about bioactive peptides in lipid-lowering was added into Introduction section.

3) The author introduced the research progress of antihypertensive peptide from these three aspects in detail, but what are the problems at present, such as whether to carry out animal experiment verification, etc. Most IC50 measurements are only conducted through in vitro experiments. Have animal experiments and population studies been carried out? What are the specific results?

Most of the publications concern the results of in vitro tests. Clinical and animal studies are few. Information on the results of in vivo studies has been added.

4) in the section “Peptides against obesity and insulin resistance”, We haven't seen the research results of peptide related to weight loss.

The research results about anti-obesity peptides were added.

5) about the reference 30: Antioxidant is not equal to anti-inflammatory, and therefore the author needs to carefully organize this part.

Thank you very much for your valuable comment. The title of the section was inadequate to the content of the item.

6) in the section “Diet factors vs. metabolic syndrome”, It is necessary to emphasize the possible role of legumes in diet, especially the protein hydrolysates of legumes, and not deviate from the theme of this review.

The role of legume in diet for people suffer from metabolic syndrome was added.

7) Better add the prospect research in the end of the review.

The prospect research in the end of the manuscript was added.

Round 2

Reviewer 2 Report

The responses and the changes made to the article are appropriate. I recommend that this paper be accepted for publication.